# B-Type Fumonisins in Post-Fermented Tea: Occurrence and Consumer Dietary Exposure in Guangxi, China

**DOI:** 10.3390/toxins15090534

**Published:** 2023-08-30

**Authors:** Taotao Qiu, Jialin Zhu, Huayi Zhang, Biyun Xu, Yanju Guo, Jingrong Li, Xin Xu, Fenglin Peng, Weiguo Liu, Shengmei Zhao, Zuocheng Yin, Shihong Mao

**Affiliations:** 1College of Physical Education and Health, Guangxi Normal University, Guilin 541004, China; gxqiutaotao@163.com (T.Q.); shengmeizhao123@163.com (S.Z.);; 2College of Tourism & Landscape Architecture, Guilin University of Technology, Guilin 541006, China

**Keywords:** post-fermented tea, exposure assessment, risk characterization, fumonisins

## Abstract

Post-fermented tea (PFT), a commonly consumed beverage worldwide, is characterized by the rapid growth of its microbial groups and the substantial changes they undergo. Consequently, PFT may contain mycotoxins such as B-type fumonisins (FBs). This study aimed to assess the intake of FBs through the consumption of PFT among consumers in Guangxi, China. A novel quantitative method using high-performance liquid chromatography-mass spectrometry was used to determine the FB concentration in PFT products. Additionally, a PFT consumption survey was conducted using a face-to-face questionnaire, recording their body weight and PFT consumption patterns based on a three-day dietary recall method. Finally, hazard index was calculated to estimate the health risk of FBs from the consumption of PFT products in Guangxi. The results revealed that the occurrence of FBs in PFT was 20% (24/120), with a concentration ranging from 2.14 to 18.28 μg/kg. The results of the survey showed that the average daily consumption of PFT by consumers was 9.19 ± 11.14 g. The deterministic risk assessment revealed that only 0.026% of the provisional maximum tolerable daily intake of FBs was consumed through PFT, indicating that FB contamination in PFT is not a public health risk.

## 1. Introduction

Fumonisins (FUMs) are a group of mycotoxins that pose a threat to public health [1]. They are natural toxins with low molecular weight and are produced as a water-soluble secondary metabolite by *Fusarium verticillioides*, *Fusarium proliferatum*, and other *Fusarium* species, which can subsequently contaminate farm products in the field or during processing, transportation, or storage [2]. For example, *Fusarium* is the principal indigenous fungus found in the soils of subtropical tea plantations in China [3] and is a potential source of contamination in the production of tea. There are four categories of FUMs: A, B, C, and P. Notably, the B-type fumonisins (FBs) are the most abundant and toxic family, including fumonisin B_1_ (FB_1_), fumonisin B_2_ (FB_2_), and fumonisin B_3_ (FB_3_). The occurrence of FBs is often directly related to climatic conditions. Guangxi, one of the cities in China where post-fermented tea (PFT) is commonly consumed, has a typical subtropical monsoon climate with short winters, long summers, and abundant annual precipitation. The climatic conditions in Guangxi are extremely favorable for the growth of toxigenic fungi. Moreover, FBs are mainly found in cereals and other agricultural products and have many toxic effects, such as carcinogenicity, immunotoxicity, teratogenicity, and hepatotoxicity, as demonstrated through tests conducted on rodents [4,5]. In addition, FB_1_ has been classified as belonging to the International Agency for Research on Cancer (IARC) Group 2B substances, implying that this mycotoxin is a possible human carcinogen [6] and has been shown to induce human esophageal cancer in many parts of Central America, Asia, and South Africa [7,8]. Additionally, FB_2_ and FB_3_ are almost structurally identical to FB_1_ at a molecular level; therefore, the magnitude of their toxicity may be at the same level as that of FB_1_ [9]. Due to their toxicity and risk to human health, the Joint Expert Committee on Food Additives evaluated FBs and allocated this mycotoxin a provisional maximum tolerable daily intake (PMTDI) of 2 μg/kg·bw/day [10]. Therefore, the need to detect mycotoxin contamination in food and assess their exposure to the human population is increasingly necessary for safeguarding human health.

As a healthy beverage known for its characteristic flavor, PFT, including the Pu-erh, Liupao, and Fu brick varieties, is a staple beverage that is consumed daily worldwide [11]. The components of PFT have unique biological activities, including antioxidant, antibacterial, antimutagenic, and anti-cancer properties, as well as health benefits that may aid in the prevention of cardiovascular disease [12]. The long aging process and pile fermentation method that is integral to the production of PFT are key to the formation of its characteristic flavors and active ingredients and involve the rapid growth of and substantial changes in the microbial groups present in the tea. Microbial groups can promote the transformation of the original tea compounds, improve the flavor of tea, and generate new active ingredients, including polyphenols, caffeine, and aroma-forming substances [13]. However, compared with the aging conditions of raw Pu-erh tea produced from non-fermented tea leaves, pile fermentation can enhance the formation of mycotoxins [14]. In addition, FB contamination can occur at any stage of commodity production, especially under warm and humid climatic environmental conditions [15]. Notably, PFT is packaged in various forms, such as baskets, cans, paper packaging, or aluminum foil bags, in Guangxi, China. Poor storage and processing conditions may lead to the growth of toxigenic microfungi, particularly under high temperature and humidity conditions. The presence of toxigenic microfungi in PFT is a risk factor for FB contamination. Furthermore, considering the water solubility of FBs, PFT consumption may be an important factor in the exposure of tea consumers to FBs. Considering the health risks associated with FBs, the risk assessment of PFT consumption has become increasingly important.

Consequently, in the last decade, the risk assessment of mycotoxin exposure from PFT consumption has received heightened attention. Studies on the presence of mycotoxins in PFT have been conducted in several countries. For example, a preliminary study on mycotoxins in PFT reported that all 36 Pu-erh tea samples purchased from Italian markets tested positive for aflatoxins (AFs) and ochratoxin A (OTA) [16]. In a recent study conducted in China, OTA and zearalenone were detected in PFT samples [17,18]. In another study, the mycotoxin citrinin was detected in Chinese Liupao tea [19]. However, FB contamination of PFT has not been widely studied in China, and there are gaps in the existing literature related to FB residues and the risk assessment of PFT consumption. 

To avoid the toxic effects of FB exposure, an accurate and specific mycotoxin detection method is critical for evaluating the safety of tea samples. In the current study, high-performance liquid chromatography-tandem mass spectrometry (HPLC-MS/MS) was the preferred method for detecting mycotoxins due to its high accuracy, stability, and reproducibility. Currently, HPLC-MS/MS is widely used for the detection of FBs in human food and animal feed [20,21,22,23]. Moreover, considering the unique mycotoxin profile of PFT products compared with those of other food products and animal feed, a quantitative method of FB detection with high accuracy and broad applicability is required. Additionally, we calculated the dietary risk of FB exposure by comparing the estimated daily intake (EDI) with the PMTDI. The EDI comprises a combination of food consumption data, typical contamination levels, and consumer body weight. Although previous surveys have shown that FB exposure levels in many countries, including Tanzania, Somalia, Mexico, and Brazil, exceeded the PMTDI [24,25,26,27], indicating potential health risks from dietary exposure to different food products, few studies have assessed the potential human health risks of consuming PFT contaminated with FBs. To the best of our knowledge, the risk assessment of FBs in PFT in Guangxi has not been previously reported. Therefore, the aims of this study were to develop a detection method to monitor FBs in PFT, assess the occurrence of FBs in PFT in this geographical region for the first time, and conduct a risk assessment of FBs exposure for PFT consumers in Guangxi. We hypothesize that the FBs contaminating PFT are completely ingested by tea consumers.

## 2. Results and Discussion

### 2.1. Method Development and Validation

Post-fermented tea products are rich in fiber, pigments, polyphenols, and caffeine, which are easily coextracted with mycotoxins and strongly influence the accurate quantitation of FBs. Conventionally, an acetonitrile-water mixture (at different percentages) is used as the extraction solvent for mycotoxin analysis [20]. Acetonitrile, as a polar extraction solvent, can reduce the extraction of pigments and lipophilic materials such as chlorophylls and fats and has a high capacity to extract plant ingredients with different polarities [28]. Furthermore, acidified solutions have been shown to reduce the interference of proteins and sugars in mycotoxin extraction [29], and acetic acid has been widely used for the extraction of FBs [30,31]. Moreover, FBs are hydrophilic mycotoxins that are soluble in acetonitrile and water [32]. Therefore, in this study, an acetonitrile-water mixture (50:50, *v*/*v*) containing 5% acetic acid was used as the extraction solvent for the determination of FBs in PFT. In addition, multifunctional cleanup column (MFC) have been shown to be efficient in eliminating pigments, lipids, proteins, caffeine, and polyphenols from tea extraction solutions. Hence, an MFC was used in this study to eliminate PFT matrix interference for FB determination.

In this study, an accurate analytical method for the determination of FBs in PFT was developed using HPLC-MS/MS and validated based on recovery, linearity, limit of detection (LOD), limit of quantification (LOQ), and repeatability criteria. The retention times of FB_1_, FB_2_, and FB_3_ were found to be about 11.02 min, 12.45 min, and 11.76 min, respectively (Figure 1(B1–B3)). The HPLC-MS/MS method showed good linearity for the target FBs, with correlation coefficients ranging from 0.9989 to 0.9996, as shown in Table 1. The LOD and LOQ were calculated using the method described below. In this study, the LOD and LOQ for FBs were determined to be in the ranges of 0.7–3.0 and 2–9.0 μg/kg (Table 1), respectively. The LOD and LOQ values were lower than those previously reported, demonstrating that the HPLC-MS/MS method used in this study had a high sensitivity for FB determination [33,34]. Previous studies have used HPLC to detect FBs; however, few HPLC-MS/MS methods have been developed for FB detection in PFT [35,36]. In particular, the potential for phenolic compounds to interfere with HPLC-fluorescence or HPLC-UV methods must be considered [18]. Therefore, we utilized an MFC to eliminate polyphenols from the tea extraction solution, thereby improving the sensitivity and accuracy of the HPLC-MS/MS method that we developed for FB determination in PFT products. 

The accuracy of the method was calculated by comparing the recovery of FBs from spiked control blanks to those of PFT samples. Each recovery experiment was carried out three times over three days, and the average recovery for FBs in the PFT samples is presented in Table 1. The mean recovery values for different FBs ranged from 67.7 to 77.13%, and the RSD was 5.81%–7.9%. These results indicate that the PFT exhibited the strongest matrix effect, which is consistent with the results of previous studies [37]. Furthermore, the recovery values for each FB determined by the newly developed method satisfied the analysis requirements of Commission Regulation (EC) No. 401/2006 (recovery values: 60–120%, RSD ≤ 30%) [38].

### 2.2. Occurrence and Concentration of FBs in PFT 

The newly developed HPLC-MS/MS method was applied to analyze 120 commercial PFT samples, including Pu-erh, Liupao, and Fu brick teas. The HPLC-MS/MS chromatograms of Liupao tea contaminated with FB_1_ and FB_2_ and a blank sample without FBs contamination are shown in Figure 1(C1,C2,D). The present study revealed an incidence of positive samples of 20.00% (24/120), with levels of FB_1_ and FB_2_ contamination in the range of 2.00–18.00 μg/kg (Table 2). Of these, 80% of samples showed FB concentrations lower than the LOD, 10% of samples were contaminated with FB_1_, 16.67% of samples were contaminated with FB_2_, 6.67% of samples were co-contaminated with FB_1_ and FB_2_, and FB_3_ was not detected in any sample. Notably, fungal hyphae were not observed in the positive samples by direct visual inspection, even when using optical microscopy (Figure 1(A1,A2)). Therefore, it would be difficult for consumers to assess FB contamination in PFT with direct visual inspection when buying PFT products. The mean FB_1_ of all positive samples and total samples investigated was 5.42 and 0.54 μg/kg, respectively, and the mean FB_2_ of all positive samples and total samples investigated was 5.70 and 0.95 μg/kg, respectively. The results demonstrated that the highest levels of FB_1_ and FB_2_ contamination were found in Fu brick tea at 11.81 and 18.28 μg/kg, respectively. Meanwhile, Liupao and Pu-erh teas showed the highest levels of FB_1_ contamination at 7.07 and 10.44 μg/kg, respectively, and FB_2_ contamination at 11.00 and 15.00 μg/kg, respectively. The lack of detectable FB_3_ content in the 120 PFT samples analyzed was in accordance with results reported in several other studies on Pu-erh, green, and black teas [39,40]. However, the FB_1_ and FB_2_ values in this study were higher than those found in Pu-erh tea by Haas et al. [16] and PFT by Ye et al. [18], yet they were below the detection value of black tea and medicinal plants in Portugal [41]. The differences in the levels of FB contamination may be attributed to the different sample sources, purification techniques, and quantification methods.

Several previous studies have compared FB concentrations in food products, including tea. Table 2 shows that FB_1_ and FB_2_ contamination was detected in Fu brick, Liupao, and Pu-erh teas in Guangxi, which is consistent with the Pu-erh tea contamination data of Guangzhou, China [42]. The risk of FB contamination in PFT could be due to the high temperature and humidity environmental conditions in PFT production, sales, and storage areas (Guangdong and Guangxi in China), providing an ideal condition for the growth of *Fusarium* sp. [43]. Notably, the percentage of positive samples for Liupao, Pu-erh, and Fu brick teas packaged in baskets or paper was 21.05%, 22.00%, and 21.43%, respectively. In comparison, the percentage of positive samples for Liupao and Pu-erh tea packaged in aluminum foil bags, plastic bags, or cans was 12.50% and 10.00%, respectively (Table 2). Furthermore, the presence of FB_1_ and FB_2_ concurrent contamination in Liupao, Pu-erh, and Fu brick teas packaged in baskets or paper and the mean FB_1_ and FB_2_ in both the total samples investigated and all positive samples in PFT packaged with baskets or paper were higher than those of PFT packaged with aluminum foil bags, plastic bags, or cans. These results imply that utilizing aluminum foil bags, plastic bags, and cans as packaging can effectively prevent FB contamination during PFT storage. Scant data exists regarding the mechanisms of FB contamination of PFT [15]. We theorize that FB contamination may occur in all aspects of tea planting, harvesting, production processing, distribution, and storage; however, as rapid microbial growth is a part of the lengthy aging procedure of PFT production, methods for preventing the growth of and contamination by FBs are critical.

### 2.3. Demographic Profile and PFT Consumption Patterns in Guangxi

Exposure assessment is an important yet challenging step in risk assessment and mainly describes the hazard level of a substance based on factors such as exposure routes, dose, and frequency, along with population characteristics [44]. Based on the existing literature, limited data are available on the accurate consumption of PFT in Guangxi. To obtain more accurate consumption data, average PFT product consumption data were obtained from major cities in Guangxi using a face-to-face questionnaire method (Table 3). The demographic profile shows that the male consumer population (63.18%) was higher than that of female consumers, according to the survey results presented in Table 3. The PFT consumption data were consistent with the results reported by Yao et al. [45]. Similar to our findings, Guan et al. reported that the distribution of respondents according to age showed that more than half of the participants were older than 45 years [46]. The average body weight of PFT consumers was 62 ± 11.01 kg, which aligned with the results of previous reports designating an average body weight of 60 kg for Chinese adults [47]. Moreover, we found that 55.16% of participants preferred Liupao tea, 21.65% preferred Pu-erh tea, and 18.68% typically consumed two or more types of PFT (Table 4). In terms of brewing method preference, 40.77% of consumers chose multiple brewing for preparing PFT. Table 4 displays the frequency of PFT consumption in Guangxi, which confirmed that the majority of study participants drink PFT 1–7 times a week. In addition, Figure 2A shows no statistically significant differences in PFT consumption based on sex (*p* > 0.05), while significant differences were observed based on age (*p* < 0.05). The 45- to 60-year-old group showed an overall higher level of PFT consumption relative to the <45- and >60-year-old groups. The average consumption of PFT was 9.19 ± 11.14 g/day (Figure 2A), while the average consumption of PFT by unit weight was 0.15 ± 0.18 g/kg·bw/day for PFT consumers (Figure 2B). 

### 2.4. Exposure Assessment and Risk Characterization

Dietary exposure is commonly assessed using the point and probability assessment methods. Point assessment uses fixed values to determine each value in the mycotoxin assessment model, such as using the mean value to represent the average level of exposure in the population [48]. In this study, point assessment was calculated using the evaluation of the risk assessment of FBs in PFT from the chronic toxicity approach, based on methods recommended by the World Health Organization [49]. The data for levels of FB contamination and PFT consumption were combined to estimate the FB exposure level. The chronic PFT intake assessment of Guangxi consumers indicated that the level of FB exposure through the consumption of Pu-erh, Liupao, and Fu brick teas showed no observable risk (Table 5). The exposure to FBs in average PFT consumers amounted to 0.026% of the PMTDI of FBs (PMTDI, 2 μg/kg·bw/day), consumed via the teas included in this study. The average EDI of FB_1_ was 0.000274, 0.000278, and 0.000343 μg/kg·bw/day for Liupao, Pu-erh, and Fu brick teas, respectively. The average EDI of FB_2_ in this population was 0.00015, 0.000176, and 0.00032 μg/kg·bw/day for Liupao, Pu-erh, and Fu brick teas, respectively. These values are considerably lower than the PMTDI of FBs, indicating a low risk of FB toxicity for Guangxi PFT consumers. Similarly, Ye et al. detected the contaminant level of FBs in Chinese dark tea, and the hazard index (HI) values for FBs were far below 1.0 [18], indicating no risk concern for consumers. 

The maximum regulatory limits for mycotoxins in tea have been established based on residual information on dry tea. However, as water infusion is the general method of tea consumption, it is necessary to monitor mycotoxin levels in PFT, estimate the average human intake, and perform health risk assessments. Recent studies have reported that the concentration of OTA and FB_2_ in tea infusions depends on the infusion pH, and the total dissolved solids in the water do not substantially affect the transfer of mycotoxins from tea to infusions [50]. One limitation of this study was that, although FBs are water-soluble toxins, their migration rates during brewing were not addressed. Future studies are required to assess the migration rate of FBs during tea brewing to accurately determine FB intake. In this study, we hypothesized that the FBs contaminating PFT are completely ingested by tea consumers. Similar to the results of previous studies, those of the current study indicate that the actual risk of exposure to FBs through PFT consumption is lower than the estimated exposure level.

Another limitation of this study was that FB levels were only investigated in PFT. It is well known that the population of Guangxi consumes a great variety of foods on a daily basis, and the presence of contamination with multiple mycotoxins is common, especially in agricultural products [51]. One survey showed that 70% of Pu-erh tea samples tested positive for aflatoxin B_1_ (AFB_1_) [52], while another showed that AFB_2_, AFG_1_, AFG_2_, deoxynivalenol, and enniatin B_1_ were detected in tea samples [53]. Most foods, such as dairy products, cereals, beans, dried fruits, fresh fruits, and vegetable juices, are potentially contaminated with multiple mycotoxins [54,55,56]. The risk of mycotoxin contamination in human food and animal feed is widespread, owing to environmental and socioeconomic factors, agricultural practices, and production processes [24,57]. Although the results of our study show that FB contamination in PFT was not a public health risk, future studies are required to determine the combined synergistic or additive harmful effects caused by cumulative exposure to multiple mycotoxin contaminants, and an assessment of the risk of potential mycotoxin exposure from the daily intake of all foods is necessary to accurately capture the risk of intake in a population.

## 3. Conclusions

Post-fermented tea has a unique flavor and a clear bioactive effect, which are aligned with consumer preference. Consequently, the risk assessment of mycotoxin exposure in PFT has become increasingly important. A reliable and accurate method using purification steps coupled with HPLC-MS/MS was developed for determining FBs in PFT products. Overall, the incidence of FBs in the PFT samples was 20%, and the maximum level of FB_2_ was 18 μg/kg. The risk assessment shows that the EDI ranged from 0.00015 to 0.000343 μg/kg·bw/day, which was far below the PMTDI value. Therefore, FB exposure through PFT beverage consumption does not represent a health risk for Guangxi consumers. Nevertheless, considering the potential for mycotoxin contamination of multiple food and beverage products, further detection of comprehensive dietary exposure to multiple mycotoxin contaminants is necessary. This study elucidates the risk of FB hazards in PFT products sold in Guangxi and provides a reference for further research on the detection and assessment of the mycotoxin contamination of other teas in other tropical and subtropical regions. Furthermore, the study results support the implementation of local initiatives aimed at preventing mycotoxin contamination.

## 4. Materials and Methods

### 4.1. Sources of PFT Samples, Chemicals, and Reagents

A total of 120 representative commercial PFT samples (60 Pu-erh, 46 Liupao, and 14 Fu brick teas) were randomly selected and purchased from various local markets in Guangxi, China, between January and December 2021. We obtained the FB_1_, FB_2_, and FB_3_ reference standards from Pribolab Pte. Ltd. (Qingdao, China). FB_2_ and FB_3_ are mixed-standard substances. We purchased HPLC-grade methanol and acetonitrile (99% purity) from Merck (Darmstadt, Germany). All other chemicals were purchased from commercial sources. Magnesium sulfate was purchased from Shanghai Macklin Biochemical Technology Co., Ltd. (Shanghai, China) and baked at 120 °C for 12 h to remove any residual water before use. A MycoSpin^TM^ 400 MFC was purchased from Romer Labs Singapore Pte. Ltd. (Singapore). Ultrapure water was obtained using a Milli-Q system (MilliporeSigma, Billerica, MA, USA). Standard stock solutions (50 mg/L) were prepared by diluting commercial FBs in acetonitrile and water (50:50, *v*/*v*) containing 5% acetic acid. Interval standard solutions were prepared by pipetting 1.0 mL of FB_1_, FB_2_, and FB_3_ into separate ampoules. The standard stock and interval standard solutions were stored at –20 and 4 °C, respectively.

### 4.2. Sample Pre-Treatment

Acetonitrile and water (50:50, *v*/*v*) containing 5% acetic acid were selected as the extraction solvents. A MycoSpin^TM^ 400 MFC was used for purification. The detailed procedure for the sample pretreatment was as follows: the ground PFT sample (10 g) was placed in a 250 mL triangular flask, to which 100 mL of acetonitrile and water (50:50, *v*/*v*) containing 5% acetic acid were added. After 2 h of shaking followed by centrifugation, the supernatant was decanted, and 2 mL of the supernatant was placed into a 5 mL centrifuge tube containing 250 mg of magnesium sulfate. Furthermore, 100 µL of acetic acid was added, and the mixture was vortexed for 3 min. After centrifugation, 750 µL of supernatant extract was passed through the MFC for purification. After the above extract was centrifuged for 1 min at 6793× *g*, 75 µL each of supernatant and internal standard solution were added. Finally, 120 µL of the supernatant was filtered through a 0.22 μm pore membrane prior to HPLC-MS/MS analysis.

### 4.3. Sample Analysis

The FBs in the PFT samples were determined using an Agilent 1290-Sciex QTRAP 5500 HPLC-MS/MS system equipped with a Dual Agilent Jet Stream ESI source operating in the positive mode and an Agilent 7683 B autosampler. For HPLC-MS/MS analysis, FBs were separated using a Phenomenex Gemini HPLC C18 column (4.6 mm × 150 mm) (Phenomenex Inc., Torrance, CA, USA). The separation was performed at 40 °C, and the injection volume was 20 μL. The mobile phase was composed of eluent A (water with 2 mmol/L ammonium acetate and 0.5% acetic acid) and eluent B (methanol with 2 mmol/L ammonium acetate and 0.5% acetic acid), and gradient elution was as follows: 10% B for 0–1 min, 10–97% B for 13 min, 97% B over 1 min, 10–97% B over 0.1 min, and 10% B for 5 min, with a flow rate of 1 mL/min and a total run time of 20 min. For the MS analysis, the ion source temperature was 650 °C, while the collision gas was of medium strength, the air curtain gas was 35 psi, ion source gas 1 was 60 psi, and ion source gas 2 was 65 psi.

Recovery rates were evaluated by adding concentrations of FB_1_ (250 μg/kg), FB_2_ (250 μg/kg), and FB_3_ (62.50 μg/kg) into FBs-free PFT samples. Analyses were conducted once a day for three days, and recovery was determined by comparing the average peak areas of the spiked samples to those of the matrix-matched standard solutions. The LOD and LOQ were calculated based on signal-to-noise ratios (S/N) of 3 (3:1) and 10 times (10:1) the background chromatographic noise, respectively. 

### 4.4. PFT Consumption Survey

A PFT consumption survey was conducted in Guangxi, China, during the first half of 2021. The population sample was predetermined as PFT consumers aged 18 years or older. Participants were informed (orally and via written instructions) about the general purpose of the survey and the basic principles of anonymity, confidentiality, and data protection. The data was collected through personal interviews. As all participants were aged 18 years old or older, interviews were conducted following verbal consent. A three-section structured questionnaire was developed and used in this study. The questionnaire was designed for direct interviews according to the China Health and Nutrition Survey and general guidelines for data collection [58]. The first section included general, self-reported demographic information, including gender, age, and weight. The second section examined PFT consumption patterns, including the frequency and quantity consumed. The third section prompted respondents to report their consumption of PFT the day before and in the last three days, including the type of product and the amount of consumed products (in g). The 24 h dietary recall is the most commonly used recall method. In addition to the 24 h dietary recall, a three-day dietary recall was included to cross-check the results and capture the habitual intake of PFT. To accurately capture the quantity and type of PFT intake, the participants were provided with 5 g each of Pu-erh, Liupao, and Fu brick tea as visual aids. These products are the most commonly consumed types of PFT in Guangxi; therefore, other PFT varieties (such as Qingzhuan and Kangzhuan) were not included in this study. 

### 4.5. Exposure Assessment

An exposure assessment was developed to predict the severity and probability of adverse health effects from the intake of FBs via PFT. To characterize the FB exposure from PFT consumption, the amount of PFT consumed during a specific period and the typical FB contamination levels in PFT products were considered. In this study, the Guangxi PFT product consumption data were mathematically treated to represent the average amount (g) of PFT consumed per day based on the daily and three-day consumption reported during field research. Quantitative MS was performed to obtain preliminary estimates of the average concentration of FBs in PFT products sold in Guangxi. 

The average concentrations used in the FB exposure calculations were determined using the following formula:Ct = (M_1_ + M_2_ + … + M_n_)/*N*
where Ct is the average FB concentration, *N* is the number of testing samples, and M is the detected mean FB concentration in the PFT. The exposure to FB through PFT consumption was calculated as EDI using data on PFT consumption (μg/kg·bw/day), FB concentration, and body weight, according to the following formula:$$\mathrm{EDI}=\frac{M_{pc\times Cp}}{b\omega }$$
where *M_pc_* is the average amount of PFT consumed (g), *bw* is body weight (kg), and *C_p_* is the average concentration of FBs in PFT (μg/g). The hazard quotient (HQ) for exposure assessment was determined by comparing the EDI of FBs with the PMTDI of FBs set at 2 μg/kg·bw/day. To assess the risk resulting from dietary exposure to FBs in PFT, the HI for multiple mycotoxin exposures was calculated as the sum of the HQ using the following formula:HI = HQ_1_ + HQ_2_ + … +HQ_n_(1)

### 4.6. Statistical Analysis

Each experiment was performed in triplicate. All values are presented as the mean ± the standard error of the mean. Differences within the groups were analyzed using repeated-measures one-way analysis of variance. When the level of FBs in a PFT sample was higher than the LOD, the sample was considered positive, whereas an FB level lower than the LOD was considered negative. The PFT samples with FB concentrations below the LOD were assumed to be equal to LOD/2 according to the EU guidelines and were used for exposure calculations [59]. IBM SPSS 19.0 software (Armonk, NY, USA) was used for statistical analysis.

## Figures and Tables

**Figure 1 toxins-15-00534-f001:**
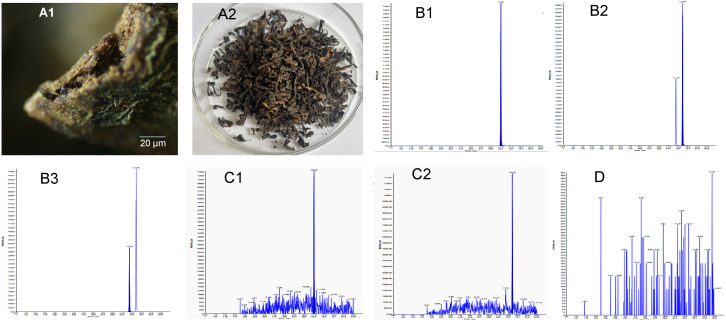
The microscope observation (**A1**) and photograph (**A2**) of the PFT samples; HPLC-MS/MS chromatogram of the FB_1_ (**B1**), FB_2_ (**B2**), and FB_3_ (**B3**); Total ion chromatogram of PFT products contaminated Liupao tea with FB_1_ (**C1**) and FB_2_ (**C2**); a blank Liupao tea (**D**).

**Figure 2 toxins-15-00534-f002:**
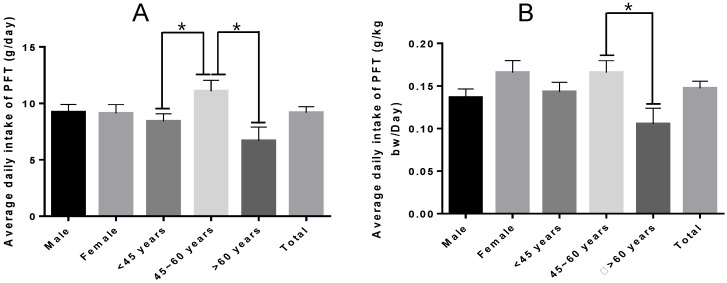
Average daily intake of PFT (**A**) and average daily intake of PFT by unit weight (**B**) in Guangxi (one-way ANOVA). * Mean significant difference (*p* < 0.05).

**Table 1 toxins-15-00534-t001:** Average recoveries, relative standard deviations (RSDs), limit of detection (LOD), limit of quantification (LOQ), linear regression equations, correlation coefficients (R^2^) for FB determination by HPLC-MS/MS in PFT.

Mycotoxins	Fortification Level (μg/kg)	Recovery (%)	RSD (%)	LOQ (μg/kg)	LOD (μg/kg)	Regression Equation	R^2^
Fumonisin B_1_	250	67.7	5.81	3	9	y = 11,020x + 4705.7	0.9994
Fumonisin B_2_	250	76.47	7.9	0.7	2	y = 20,797x + 116,020	0.9989
Fumonisin B_3_	62.5	77.13	5.78	0.7	2	y = 26,700x + 53,722	0.9996

**Table 2 toxins-15-00534-t002:** Occurrence and concentrations of FBs in PFT products sold in Guangxi.

PFT Products	Package Type	Positive/Total	Positive (%)	Number of Samples					FB_1_ Maximum (μg/kg)	FB_2_ Maximum (μg/kg)	FB_1_ M1 (μg/kg)	FB_1_ M2 (μg/kg)	FB_2_ M1 (μg/kg)	FB_2_ M2 (μg/kg)
				<LOD (μg/kg)	FB_1_	FB_2_	FB_3_	FB_1_ and FB_2_						
Liupao teas	Basket packing, and paper packaging	8/38	21.05	30	4	8	0	4	7.07 ± 0.24	11.00 ± 0.32	0.55	5.25	0.87	4.13
	Aluminum foil bag packaging, plastic bag packaging, and canned	1/8	12.5	7	0	1	0	0	/	3.11 ± 0.14	/	/	0.39	3.11
Pu-erh teas	Basket packing, and paper packaging	11/50	22	39	6	7	0	2	10.44 ± 0.11	15.00 ± 0.68	0.6	5	0.92	6.57
	Aluminum foil bag packaging, plastic bag packaging, and canned	1/10	10	9	0	1	0	0	/	6.04 ± 0.78	/	/	0.6	6.04
Fu brick teas	Basket packing, and paper packaging	3/14	21.43	11	2	3	0	2	11.81 ± 0.54	18.28 ± 0.34	1	7	1.86	8.67
Total		24/120	20	96	12	20	0	8	11.81 ± 0.54	18.28 ± 0.34	0.54	5.42	0.95	5.7

/: not detected. M1: mean value of total samples. M2: mean value of all positive samples.

**Table 3 toxins-15-00534-t003:** Demographic profile of PFT consumption participants sampled in Guangxi (N = 910).

Demographic Characteristics		Population Ratio
Gender	Male	575 (63.18%)
	Female	335 (36.81%)
Age	18–44	372 (40.88%)
	45–60	437 (48.02%)
	>60	101 (11.10%)
Respondents distribution	Nanning	96 (10.55%)
	Liuzhou	174 (19.12%)
	Guilin	228 (25.06%)
	Wuzhou	100 (10.99%)
	Other cities	312 (34.36%)
Total		910 (100%)
Average body weight (kg)		62.11 ± 11.01

**Table 4 toxins-15-00534-t004:** Preferences and frequency of PFT consumption in Guangxi. (N = 910).

	PFT Products	Sample Population Ratio		Brewing Method	Sample Population Ratio
Tea preference	Liupao teas	502 (55.16%)	Brewing method	Multiple brewing for drinking	371 (40.77%)
	Pu-erh teas	197 (21.65%)	preference	Single brewing for drinking	338 (36.70%)
	Fu brick teas	41 (4.51%)		Long-term steaming for drinking	141 (15.49%)
	More than one type of tea	170 (18.68%)		More than one brewing method	64 (7.03%)
Times/day	Sample population ratio	Times/week	Sample population ratio	Times/month	Sample population ratio
1–2	639 (70.22%)	1–7	656 (72.09%)	1–15	398 (43.74%)
3–4	252 (27.70%)	8–14	197 (21.65%)	16–30	377 (41.43%)
>4	19 (2.09%)	>14	57 (6.26%)	>30	135 (14.84%)

**Table 5 toxins-15-00534-t005:** Dietary exposure assessment of FB consumption for PFT consumers in Guangxi.

PFT Products	FB Mean		FB Intake		HQ_1_ (%)	HQ_2_ (%)	HI (%)
	FB_1_ Mean (μg/kg)	FB_2_ Mean (μg/kg)	FB_1_ Intake (μg/kg·bw/day)	FB_2_ Intake (μg/kg·bw/day)			
Liupao tea	1.83	1	0.000274	0.00015	0.0137	0.00749	0.02119
Pu-erh tea	1.85	1.17	0.000278	0.000176	0.01388	0.00878	0.02265
Fu brick tea	2.29	2.13	0.000343	0.00032	0.01714	0.01599	0.03313
Average	1.99	1.43	0.000298	0.000215	0.0149	0.01075	0.02566

FBs mean: FB mean value of total samples. PMTDI = 2 μg/kg·bw/day. Average body weight = 62.11 kg. FB intake = (FB mean × PFT consumption)/average body weight. HQ = FB intake × 100/PMTDI. HI = HQ_1_ + HQ_2_.

## Data Availability

The data presented in this study are available on request from the corresponding author.

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
