# Peer review of "B-Type Fumonisins in Post-Fermented Tea: Occurrence and Consumer Dietary Exposure in Guangxi, China"

_toxins, 2023, doi:10.3390/toxins15090534_

Round 1

Reviewer 1 Report

The article presents a method for detecting FBs in post-fermented tea (PFT) using HPLC-MS/MS and assesses the risk of FBs in PFT in Guangxi region. The study also includes a survey of PFT consumption and preferences in the region to further analyze the potential risks of FBs. While the research is of certain significance and value, there are some areas that need improvement. The content of the study is relatively limited, and the analysis of the survey questionnaire occupies a significant portion, which may not be suitable for publication as a full research article. Instead, it could be considered for revision and publication as a short communication. Additionally, there are other issues that require clarification and revision.

Introduction:

(1) It is worth considering why the authors only established a detection method for FBs and not for other compounds in the fumonisin family. Moreover, has the author considered designing a method to detect metabolites of common fungi that may contaminate post-fermented tea? This would be more meaningful than solely focusing on FBs.

(2) Could the Introduction provide information on common fungi and their metabolites that contaminate post-fermented tea? Also, why did the author choose to establish a detection method for FBs? Are FBs one of the main contaminants in post-fermented tea?

(3) In Line 108, while the study evaluates the risk of FBs in post-fermented tea in Guangxi province, this should not be the main objective of the study. The author should optimize the description of this objective and extend it to predict and assess the contamination status of similar teas in other regions.

Results and Discussion:

(1) In Fig. 1B2 and B3, as FB2 and FB3 were individually added as standard substances for detection, why do two distinct peaks appear? The author should explain the reason.

(2) Earlier, the author mentioned that the chemical structures of FB1, FB2, and FB3 are similar, but why are the recovery rates of FB1 significantly lower, and the quantitation and detection limits significantly higher than FB2 and FB3 in the established HPLC-MS/MS method? This needs clarification.

(3) The term "positive samples" mentioned in the text lacks explanation. It would be helpful if the author could define "positive samples" and clarify whether they include samples that tested positive for FBs but were below the quantitation limit. The content of Table 2 can be confusing, and the author should provide a detailed explanation in the text. Moreover, in Table 2, the author should ensure that each word is on the same line.

(4) Regarding the data source for Table 3, the author should provide the original files as supplementary material for upload.

(5) The study mentions limitations such as variations in toxin content before and after tea infusion and the potential effects of infusion time on toxin dissolution. It also highlights that only FBs in tea were detected without exploring other foods. These limitations are significant and should be addressed in future research. For instance, if the study aims to detect FBs in other foods, further optimization of the detection method may be required to improve recovery rates and quantitation limits. It is recommended that the author address these aspects in future research.

Minor editing of English language required

Reviewer 2 Report

The manuscript assessed the exposure of population in Guangxi to fumonisins through the consumption of post-fermented tea as well as the associated health risks. The work described in this manuscript was well conducted. The analytical method used for the determination of fumonisins was validated with important parameters such as recoveries within the acceptable value. Furthermore, the exposure assessment and estimation of hazard index value to estimate the health risk of fumonisins following the consumption of the fermented tea products were very well conducted. A number of limitations associated with the study, which this reviewer would have raised, have been outlined by the authors.

Few minor points to consider:

The abbreviations, FBs and FB were used inconsistently throughout the paper

Line 42-43, other factors such as agricultural practice also have direct impact on mycotoxin contamination.

English language is fine

Reviewer 3 Report

The authors evaluated the occurrence and dietary exposure assessment of fumonisin B1, 2, and 3 through consumption of fermented tea in Guangxi, China

Major comments

The abstract should be a total of about 200 words maximum. The authors should summarise the abstract. It should be written in a concise and coherent manner using the following pattern - background, method, result, and conclusion (refer to guideline for authors)

Comments

Line 6, “…beverages all of world…”. correct sentence

Line 16-17, restructure the sentence

Line 101, change “has” to ”is”

Line 107-109, correct sentence

 Moderate editing of English language required

Round 2

Reviewer 1 Report

Dear Author

Thank you for your prompt response and the thorough revisions you've made in line with the feedback provided. Your responses effectively addressed the concerns raised and your manuscript modifications reflect careful attention to detail.

I also noted your mention of completed preliminary experiments. It might be beneficial to include these results as supplementary material to provide readers with a comprehensive perspective on the research foundation.

Your diligence in refining the manuscript is truly appreciated. Feel free to contact me if you need further assistance or have any questions.